# Application of Adaptive Weighted Strong Tracking Unscented Kalman Filter in Non-Cooperative Maneuvering Target Tracking

Pu Huang [1,2,†], Hengnian Li [2,†], Guangwei Wen [1,†] and Zhaokui Wang [1,*,†]

1   School of Aerospace Engineering, Tsinghua University, Beijing 100084, China
2   State Key Laboratory of Astronautic Dynamics, Xi'an 710043, China
*   Correspondence: wangzk@tsinghua.edu.cn
†   These authors contributed equally to this work.

**Abstract:** An adaptive weighted strong tracking unscented Kalman filter is proposed in this paper for long-range relative navigation alongside non-cooperative maneuvering targets. First, an equation for obtaining the relative motion of two bodies is derived, it can be well adapted for a problem that has medium or long-distance. Secondly, a variance statistics function is introduced in the method to calculate residual weight in real time. The residual weight can be used to adjust the contribution of different measurement information to the fading factor. In this way, the sensitivity of the system to small pulse maneuvers is improved. Finally, the mean and covariance of the posterior state are calculated by the unscented transformation. A replacement equation for the fading factor is derived to improve the first-order approximation accuracy for a strong tracking system. Impulsive maneuvers with three different magnitudes are employed in a series of tests. Results from different methods showed that the proposed method could effectively detect pulse maneuvers with low latency. The proposed method is also numerically stable.

**Keywords:** adaptive weighted; relative navigation; fading factor; unscented transformation

## 1. Introduction

In recent years, as the number of spacecraft has increased drastically, space situation awareness (SSA) technology has attracted more and more attention [1–3]. In particular, a real-time orbit determination method for non-cooperative maneuvering targets is urgently needed for space security [4–7]. For a ground system, the response time and the accuracy of orbit determination no longer meet the requirements of relative navigation with targets that are 100 km apart. To obtain space situational awareness ability, it is vital to develop an orbit determination method for medium or long-distance space non-cooperative maneuvering targets based on a spaceborne platform [8]. For the current research, methods for maneuvering detection and relative navigation for non-cooperative targets are investigated [9].

In order to accurately detect the maneuvers of a non-cooperative target, the system state equation and measurement equation are needed, and a filtering method needs to be appropriately selected [10–13]. The system state equations that describe the relative navigation are composed of an algebraic method and a geometric method. Based on relative motion between spacecraft, the algebraic method establishes simplified dynamic equations [14], which include the Clohessy–Wiltshire (CW) equation, the Tschauner–Hempel (TH) equation, and the relative motion equation, which considers the perturbation term. On the other hand, the algebraic method can directly be employed for short-range missions such as satellite formation and spacecraft rendezvous and docking. However, because of the assumptions that were made beforehand, the algebraic method is not suitable for long-range (more than 100 km) relative navigation. The geometric method describes the





relative motion of two spacecraft by orbital elements. Perturbations on the spacecraft can be described clearly in the geometric method, and the orbital design can be easily conducted [15–17]. Nevertheless, the equations for the geometric method are complex, and it is difficult to be applied in the relative navigation for a non-cooperative target. The double line-of-sight angle measurement method and line-of-sight angle combined-ranging method are usually employed in the measurement equation [18–21]. For the double line-of-sight angle measurement method, two observation satellites are required to observe the target satellite at the same time. The cost of the method is high, and there are lots of risks. The line-of-sight angle combined-ranging method is generally utilized in long-distance observation. For this method, the relative distance, velocity and relative angle can be obtained by different sensors, which has the advantages of low cost and strong reliability. It also meets the requirements of relative navigation. The filtering method generally adopts the non-linear Kalman filtering method [22–24], including extended Kalman filtering (EKF) and unscented Kalman filtering (UKF). However, when the target is being maneuvered, especially a non-cooperative target, the lack of maneuvering information will cause disagreement between the dynamic model and the actual maneuvering state, which affects the tracking performance of the filter, and can even lead to divergence. To solve the issue, Zhou proposed a strong tracking method and employed a residual orthogonality principle to make the system more robust to maneuverings [25,26]. Wang combined the strong tracking filter (STF) with the unscented Kalman filter (UKF), and they proposed the strong tracking unscented Kalman filter (STUKF) method to improve the non-linear processing ability of the system [27]. Jiang proposed residual normalized strong tracking extended Kalman filter, which adjusts the fade factor by residual normalization to improve the tracking ability of the filter [28]. However, for a strong non-linear system, with small maneuver changes, detection latency and filtering divergence are presented in the existing methods.

To solve the abovementioned problems, the adaptive weighted strong tracking unscented Kalman filter (AWSTUKF) method is proposed in the present paper. First, the relative motion equation that is suitable for a medium or long-distance problem is derived from the relative motion equation of a space target. Secondly, considering that a small pulse maneuver is hard to be detected by a strong tracking filter, the proposed method introduces the statistical variance function, which calculates a weight coefficient in real-time. The weight coefficient controls the contribution of different measurement information to the attenuation factor. Therefore, the filter can detect the small pulse maneuver faster with higher precision. Finally, the mean and covariance of the posterior state are calculated by using the unscented transformation to overcome the problem of first-order low-approximation accuracy in a strong tracking filter system. Impulsive maneuvers with three different magnitudes are employed in a series of simulations. The result showed that the proposed method could effectively detect pulse maneuvers with low delay. It also has a good tracking performance and good numerical stability.

The remainder of the manuscript is organized as follows. In Section 2, the dynamic model and the observation equation are introduced. Details of the proposed AWSTUKF method are presented. In Section 3, different simulation scenarios are designed to illustrate the feasibility of this method with respect to various maneuvers. The performance of the proposed method is verified by comparing the CWSTF and the STUKF methods. In Section 4, conclusions and discussions are provided.

## 2. Methodology

### 2.1. Relative Navigation Dynamics Equation and Observation Equation

To describe the relative position and velocity of two spatial objects, a reference coordinate system is first defined as the orbital coordinate system (Figure 1). The earth center, $O_E$, is the origin. The origin of the orbital coordinate system is the centroid of the space-based platform. The $X$-axis is in the direction of the earth center to the space-based platform. $Y$-axis points in the velocity direction of the space-based platform, and it is perpendicular to the $X$-axis on the orbital plane. The $Z$-axis points in the normal direction of the orbital

plane. The coordinate system is a right-hand system. A typical relative position of the space-based platforms and the non-cooperative targets are plotted in Figure 2.

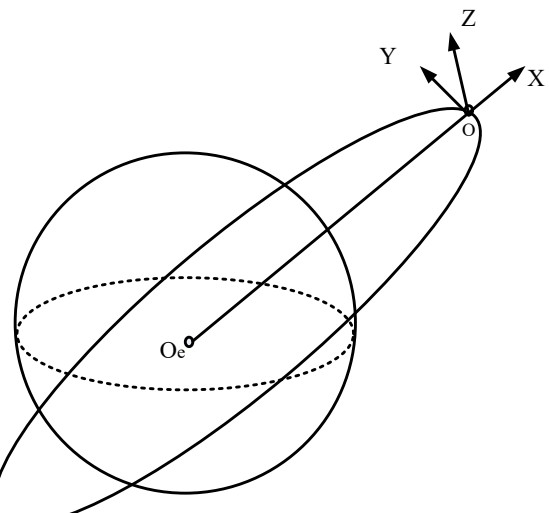

**Figure 1.** The Orbital Coordinate System.

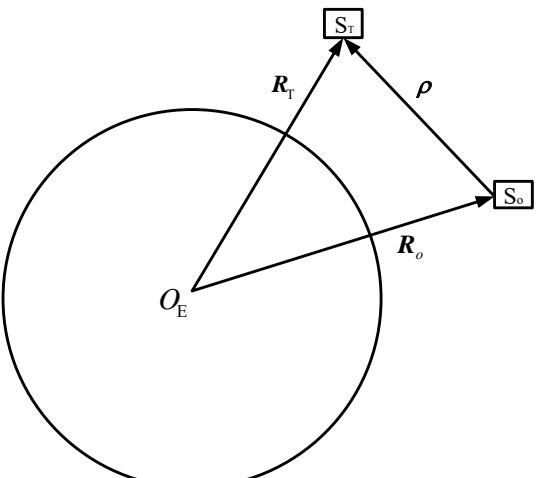

**Figure 2.** The Position Vector from the Observer to the Target.

In Figure 2, $S_o$ is the space-based platform or the observer, $S_T$ is the non-cooperative target, $R_o$ and $R_T$ are the position vectors of the space-based platform and the non-cooperative target, respectively. The relative position vector is defined as $\rho = R_T - R_o$. According to the vector differential method in the dynamic coordinate system, the second-order derivative of the relative position vector is,

$$\frac{\mathrm{d}^2\rho}{\mathrm{d}t^2} = \ddot{\rho} + 2\omega_o \times \dot{\rho} + \omega_o \times (\omega_o \times \rho) + \dot{\omega}_o \times \rho, \tag{1}$$

where $\frac{\mathrm{d}^2\rho}{\mathrm{d}t^2}$ is the absolute derivative, $\omega_o = [0, 0, n]^T$ is the orbital angular velocity vector of the space-based platform, $\dot{\rho}$ and $\ddot{\rho}$ is the local derivative in the dynamic coordinate system. Assuming that the spacecraft is not maneuvered,

$$\frac{\mathrm{d}^2\rho}{\mathrm{d}t^2} = \Delta\rho_\mathrm{g} + \Delta\rho_\mathrm{P}. \tag{2}$$

In Equation (2), $\Delta\rho_g$ is the central gravitational acceleration, and $\Delta\rho_P$ is the perturbation acceleration. Combining Equations (1) and (2), neglecting perturbation acceleration, two-body relative motion equations can be obtained as,

$$\ddot{\rho} = -2\omega_o \times \dot{\rho} - \omega_o \times (\omega_o \times \rho) - \dot{\omega}_o \times \rho + \Delta\rho_g \tag{3}$$

The relative vector of the two targets is projected to the orbital coordinate system. The gravitational acceleration of the non-cooperative target and the space-based platform in the orbital coordinate system can be derived as,

$$a_T = -\frac{\mu}{r_T^3}[r_o + x, y, z]^T, \tag{4}$$

$$a_o = -\frac{\mu}{r_o^3}[1, 0, 0]^T, \tag{5}$$

where $\mu$ is the gravitational constant of the earth, $r_T$ is the center distance of the non-cooperative target, and $r_o$ is the center distance of the space-based platform. The central gravitational acceleration difference between the two is calculated as,

$$\Delta\rho_g = -\frac{\mu}{r_T^3}[r_o + x, y, z]^T + \frac{\mu}{r_o^3}[1, 0, 0]^T, \tag{6}$$

where $r_T = \sqrt{(r_o + x)^2 + y^2 + z^2}$. Substituting Equation (6) into Equation (3) and replace it into the coordinate component form, the system state equation can be obtained as,

$$\dot{X} = \begin{bmatrix} \dot{x} \\ \dot{y} \\ \dot{z} \\ n^2 x + 2n\dot{y} + \mu/r_o^2 - \mu(r_o + x)/r_T^3 \\ n^2 y - 2n\dot{x} - \mu y/r_T^3 \\ -\mu z/r_T^3 \end{bmatrix} + w, \tag{7}$$

where,

$$X = [x, y, z, \dot{x}, \dot{y}, \dot{z}]^T,$$

is the state vector of the filter, $n$ is the rotational angular velocity of the earth, $w$ represents the model error which can be represented by Gaussian white noise. This equation describes the relative motion between the target spacecraft and the space-based platform. In the equation, only gravity of the earth's center is considered, it does not consider the earth's non-spherical perturbation, atmospheric drag, and unknown maneuvers. Different from the CW equation, the equation considers the influence of the relative distance between targets on the state equation, which is more suitable for medium and long-distance relative navigation systems [8].

In the relative navigation observation system, the main measurement equipment are line-of-sight angle measurement and range measurement. In order to simplify the calculation, the measurement coordinate system and the orbital coordinate system are overlapped, and the azimuth and elevation angles are defined as [20,21],

$$\left.\begin{array}{l} A = \arctan\left(\frac{x}{y}\right) \\ E = \arctan\left(\frac{z}{\sqrt{x^2+y^2}}\right) \end{array}\right\}. \tag{8}$$

The range and range rate in the target coordinate measurement system can be expressed as [24],

$$\left.\begin{array}{l} r_{oT} = \sqrt{x^2 + y^2 + z^2} \\ \dot{r}_{oT} = \frac{dot(r_{oT}, \dot{r}_{oT})}{r_{oT}} \end{array}\right\}, \tag{9}$$

Hence, the observation equation of the relative navigation system can be established as,

$$
Z = \begin{bmatrix} \sqrt{x^2 + y^2 + z^2} \\ \arctan\left(\frac{x}{y}\right) \\ \arctan\left(\frac{z}{\sqrt{x^2+y^2}}\right) \\ \frac{dot(r_{oT}, \dot{r}_{oT})}{r_{oT}} \end{bmatrix} + v,
\tag{10}
$$

where $v$ represents the measurement error which is analogous to Gaussian white noise, the state equation and observation equation for non-cooperative target tracking are included in Equations (7) and (10). It can be observed that the state function and observation function are non-linear. To solve the equation, the non-linear state estimation method should be employed.

*2.2. Adaptive Weighted Strong Tracking Unscented Kalman Filter*
2.2.1. Strong Tracking Filter

For the strong tracking filter (STF), it is basically an extended Kalman filter. By utilizing fading factor $\lambda_{k+1}$, the one-step prediction covariance matrix $P_{k+1|k}$ is corrected in real-time. The gain of the filtering $K_{k+1}$ is adjusted to satisfy [25,26],

(1) $\quad E\left[(x_{k+1} - \hat{x}_{k+1})(x_{k+1} - \hat{x}_{k+1})^T\right] = \min,$

(2) $\quad E\left[\gamma_{k+1+j}\gamma_{k+1}^T\right] = 0, k = 0, 1, \cdots, j = 1, 2, \cdots.$

Condition (1) is an effectiveness indicator of the EKF. Condition (2) ensures the system orthogonality on the residual output sequence. In engineering practice, as the target maneuvers, due to disagreement of the dynamic model, the estimated output of the filter will deviate from the system, resulting in a non-orthogonal residual sequence. According to this, the strong tracking filter introduces the fading factor to adjust the filtering gain online. As a result, the residual sequence remains orthogonal, and the effective information in the residual sequence is employed to maximize the tracking performance of the system. The STF is derived as,

$$
\hat{x}_{k+1|k} = F_{k+1|k}\hat{x}_{k|k},
\tag{11}
$$

$$
P_{k+1|k} = \lambda_{k+1}F_{k+1|k}P_{k|k}F_{k+1|k}^T + Q_k,
\tag{12}
$$

$$
K_{k+1} = P_{k+1|k}H_{k+1}^T\left(H_{k+1}P_{k+1|k}H_{k+1}^T + R_{k+1}\right)^{-1},
\tag{13}
$$

$$
\hat{x}_{k+1} = \hat{x}_{k+1|k} + K_{k+1}\left(y_{k+1} - \hat{y}_{k+1|k}\right),
\tag{14}
$$

$$
P_{k+1} = (I - K_{k+1}H_{k+1})P_{k+1|k},
\tag{15}
$$

where $\lambda_{k+1}$ is the fading factor, it can be determined by the orthogonality principle [11],

$$
\lambda_{k+1} = \begin{cases} \lambda_0, \lambda_0 \geq 1 \\ 1, \lambda_0 < 1, \end{cases}, \lambda_0 = \frac{tr[N_{k+1}]}{tr[M_{k+1}]},
\tag{16}
$$

$$
N_{k+1} = V_{k+1} - H_{k+1}Q_kH_{k+1}^T - R_{k+1},
\tag{17}
$$

$$
M_{k+1} = F_{k+1|k}P_{k|k}F_{k+1|k}^T H_{k+1}^T H_{k+1},
\tag{18}
$$

where $tr[\cdot]$ represents the trace of a matrix, $V_{k+1}$ is the covariance matrix of the actual output residual sequence, which can be calculated by,

$$
V_{k+1} = \begin{cases} \gamma_1\gamma_1^T, k = 0 \\ \frac{\rho V_k + \gamma_{k+1}\gamma_{k+1}^T}{1+\rho}, k \geq 1 \end{cases}
\tag{19}
$$

In Equation (19), $\rho$ is a fading factor and $0 < \rho \leq 1$. The fading factor is usually set to 0.95.

### 2.2.2. Deficiencies in the Strong Tracking Filter

From Equation (17), it can be derived that,

$$tr[N_{k+1}] = tr[V_{k+1}] - tr\left[R_{k+1} - H_{k+1}Q_k H_{k+1}^T\right]. \tag{20}$$

In Equation (20), the second term on the right-hand side of the equation is independent of the residual sequence. It can be written as $s_{k+1} = tr\left[R_{k+1} - H_{k+1}Q_k H_{k+1}^T\right]$, where $k \geq 1$, thus,

$$tr[N_{k+1}] = tr[V_{k+1}] - s_{k+1} = \frac{\rho}{1+\rho} tr[V_{k+1}] + \frac{1}{1+\rho} \sum_{i=1}^{m} \left(\gamma_{k+1}^i\right)^2 - s_{k+1}. \tag{21}$$

From Equation (16) to (18), one can observe that the denominator in $\lambda_0$ is not related to the residual $\gamma$, the size of $\lambda_0$ is mainly determined by the second term in Equation (21). However, in reality, different measurement information has significantly different values of $\gamma_{k+1}^i$. Assuming $\gamma_{k+1}^i \ll \gamma_{k+1}^j$, there are two conditions. First, the normal disturbance of $\gamma_{k+1}^j$ leads to $\lambda_{k+1} > 1$. Second, the $\gamma_{k+1}^i$ exceeds the threshold, but its influence on the $\lambda_{k+1}$ is very small, thus $\lambda_{k+1} = 1$. In other words, $\lambda_{k+1}$ is sensitive to $\gamma_{k+1}^j$, but not sensitive to $\gamma_{k+1}^i$, it will reduce the speed and accuracy of the maneuvering detection. At the same time, the Jacobi matrix $H_{k+1}$ and $F_{k+1|k}$ needs to be solved in the strong tracking system. Furthermore, there will be large deviations when the state equation is strongly non-linear. To tackle the drawbacks, in this paper, adaptive weighted strong tracking unscented Kalman filter is proposed to improve the strong tracking method.

### 2.2.3. Adaptive Weighted Strong Tracking Unscented Kalman Filter

For the contribution of different measurement information to the fading factor, the weight coefficient can be used for control and balancing. Considering that the relative measurement information mainly includes ranging, velocity measurement, azimuth angle and elevation angle, the weight coefficient is defined as,

$$w_\rho = \frac{\sigma}{\sigma_\rho}, w_A = \frac{\sigma}{\sigma_A}, w_E = \frac{\sigma}{\sigma_E}, w_{\dot\rho} = \frac{\sigma}{\sigma_{\dot\rho}}, \tag{22}$$

where $\sigma_\rho, \sigma_A, \sigma_E, \sigma_{\dot\rho}$ are the measurements for variance statistics, which can be calculated by the variance statistics function,

$$\sigma_i = \sqrt{\frac{\sum_{j=1}^{n} \left(\gamma_{k+1}^i\right)_i^2 \chi\left(\gamma_{k+1}^i, 3\sigma\right)}{\sum_{j=1}^{n} \chi\left(\gamma_{k+1}^i, 3\sigma\right)}}, i = \rho, A, E, \dot\rho, \tag{23}$$

$$\sigma = \sqrt{\frac{\sigma_\rho^2 + \sigma_A^2 + \sigma_E^2 + \sigma_{\dot\rho}^2}{4}}, \tag{24}$$

In Equation (23), $\chi\left(\gamma_{k+1}^i, 3\sigma\right)$ is the variance control function and,

$$\chi\left(\gamma_{k+1}^i, 3\sigma\right) = \left\{ \begin{array}{l} 1, \gamma_{k+1}^i < 3\sigma \\ 0, \gamma_{k+1}^i > 3\sigma \end{array} \right., i = \rho, A, E, \dot\rho. \tag{25}$$

In the filtering process, the residual information is recorded in real-time. When the residual exceeds the threshold, the outliers can be considered and eliminated. The amount

of statistical data can be adjusted by simulation. The initial weight coefficient can be set as the initial variance provided by the measurement equipment. Let $\gamma'_{k+1} = \eta \cdot \gamma_{k+1}$, where, $\eta$, is a diagonal matrix and $\eta = diag(w_\rho, w_A, w_E, w_{\dot\rho})$. Therefore, Equation (19) becomes,

$$V'_{k+1} = E\left[\gamma'_{k+1}\left(\gamma'_{k+1}\right)^T\right] = H_{k+1}P_{k+1|k}H_{k+1}^T + R_{k+1}, \tag{26}$$

where $V'_{k+1} = \eta V_{k+1}\eta^T$. When the target maneuvers, the system output residual is no longer similar to the Gaussian white noise.

$$\eta V_{k+1}\eta^T > H_{k+1}P_{k+1|k}H_{k+1}^T + R_{k+1} \tag{27}$$

By substituting Equation (10) into Equation (27), the forced output residual is analogous to Gaussian white noise again and,

$$\eta V_{k+1}\eta^T = H_{k+1}\left(\lambda_{k+1}F_{k+1|k}P_{k|k}F_{k+1|k}^T + Q_k\right)H_{k+1}^T + R_{k+1}, \tag{28}$$

Equation (28) becomes,

$$N_{k+1} = \lambda_{k+1}M_{k+1}, \tag{29}$$

where $N_{k+1} = \eta V_{k+1}\eta^T - R_{k+1} - H_{k+1}Q_kH_{k+1}^T$ and $M_{k+1} = H_{k+1}F_{k+1|k}P_kF_{k+1|k}^TH_{k+1}^T$.

Since the Jacobian matrix $H_{k+1}$ and $F_{k+1|k}$ are required to solve for the $\lambda_{k+1}$ [29], it is not suitable for strong non-linear systems. The posterior mean and covariance of state can be calculated by Unscented transformation [30], the adaptive weighted strong tracking filter can also be obtained based on Unscented transformation. Before the introduction of the fading factor, the covariance matrix of state prediction is [31],

$$P_{k+1|k}^{(l)} = \sum_{i=0}^{2n} \omega_i^c \left(X_{k+1|k}^i - \hat{X}_{k+1|k}\right)\left(X_{k+1|k}^i - \hat{X}_{k+1|k}\right)^T + Q_k = F_{k+1|k}P_{k|k}F_{k+1|k}^T + Q_k, \tag{30}$$

$$P_{xy}^{(l)} = \sum_{i=0}^{2n} \omega_i^c \left(x_{k+1|k}^i - \hat{x}_{k+1|k}\right)\left(y_{k+1|k}^i - \hat{y}_{k+1|k}\right)^T = P_{k+1|k}^{(l)}H_{k+1}^T. \tag{31}$$

Assuming that $Q_k$ is a positive definite symmetric matrix, then the inverse matrix of $P_{k+1|k}^{(l)}$ must exist and can be calculated as,

$$H_{k+1} = \left(P_{xy}^{(l)}\right)^T\left(P_{k+1|k}^{(l)}\right)^{-1}. \tag{32}$$

The calculation formula of the fading factor is derived, and it can be obtained as,

$$N_{k+1} = \eta(V_{k+1})\eta - R_{k+1} - \left(P_{xy}^{(l)}\right)^T\left(P_{k+1|k}^{(l)}\right)^{-1}Q_k\left(P_{k+1|k}^{(l)}\right)^{-1}\left(P_{xy}^{(l)}\right), \tag{33}$$

$$M_{k+1} = \left(P_{xy}^{(l)}\right)^T\left(P_{k+1|k}^{(l)}\right)^{-1}\left(P_{k+1|k}^{(l)} - Q_k\right)\left(P_{k+1|k}^{(l)}\right)^{-1}\left(P_{xy}^{(l)}\right). \tag{34}$$

In summary, the Unscented transformation with higher approximation accuracy is employed to calculate the posterior mean and covariance of the state, the complexity of calculating the Jacobian matrix is reduced, and numerical stability and accuracy are improved. In addition, the weight is calculated in real-time, the contribution of residual information to the fading factor can be adjusted instantaneously, which enhances the sensitivity and robustness of the maneuver detection system.

## 3. Results

In order to examine the proposed method, simulations were conducted. The simulations were carried out in the following conditions. For the orbital parameters, the

space-based platform has an orbital height of 800 km, and it has a circular orbit with an orbital inclination of 98 degrees. The space-based observer is equipped with a lidar sensor and a visual camera. The accuracy of the distance measurement is 10 m, the accuracy of the velocity measurement is 0.1 m/s, and the accuracy of the angle measurement is 0.02 degrees. High-precision models are employed for both space-based platforms and non-cooperative targets for analysis. The initial relative state is, $X_0 = [200, 200,000, 300, -1, -10, 0.1]^T$.

The tracking begins on 26 March 2022 at 12:00:00, and it lasts for 500 s. The measurement sampling interval is 0.1 s. The initial state estimate errors are set as 200 m and 1 m/s. To demonstrate the effectiveness and benefits of the proposed method, the filtering results are compared with the class UKF method, the STUKF method and AWSTUKF in two cases.

In case 1, for the target spacecraft, the magnitude of the pulse maneuver is 5 m/s, the maneuver occurs at 300 s, and the maneuver is in the direction of its velocity. The AWSTUKF, the UKF and the STUKF methods are employed. Position estimation errors from the three methods are plotted in Figure 3, and velocity estimation errors are plotted in Figure 4. Before the pulse maneuver, the position estimation errors and the velocity estimation errors from the three methods are identical. After the maneuver, the AWSTUKF method has the fastest convergence performance, followed by the STUKF method, and the UKF method is the worst. This is because the three algorithms have different maneuvering detection delays. Due to the adaptive method, the AWSTUKF can quickly detect a maneuvering, and the fading factor is used to amplify the covariance matrix for a fast convergence of the filter. Because the STUKF does not adopt the adaptive method, the maneuvering detection delay is longer than the AWSTUKF, and the convergence also takes a longer time. The UKF does not detect pulse maneuvering, so the convergence speed is the slowest.

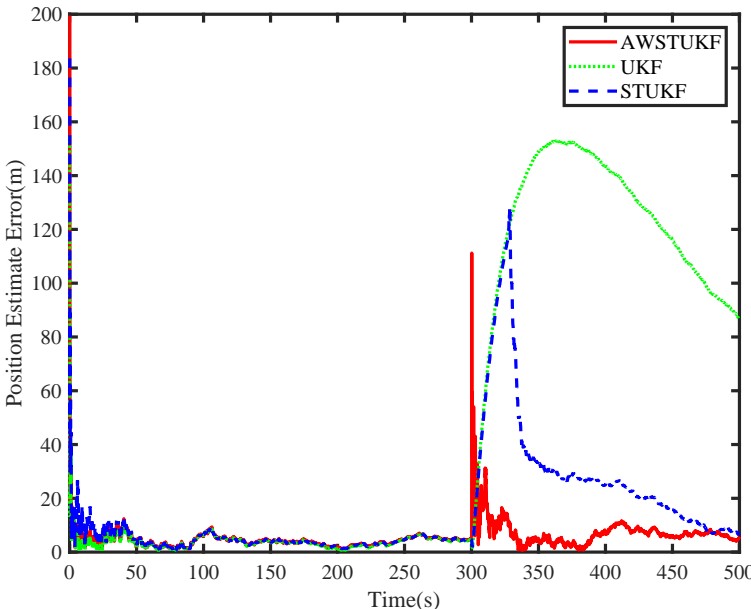

**Figure 3.** Position estimate errors of different methods.

In case 2, for the target spacecraft, the magnitude of the pulse maneuver is 0.5 m/s, which is smaller than in case 1. The purpose of this setting is to test the performance of the methods regarding a small pulse maneuver. The maneuver occurs at 300 *s*, and the pulse maneuver is along the direction of the velocity. The position estimation errors and the velocity estimation errors of the three methods are provided in Figures 5 and 6, respectively. It can be observed that the position and velocity estimation errors of the three methods are basically the same before the maneuver. After the small maneuver, the state estimate error in the proposed AWSTUKF method is reduced faster than in the other two methods. When a small pulse maneuver occurs, the STUKF fails to detect the maneuver due to the residual orthogonality is not consistent with the actual measurement, which

results in the fading factor always equal to 1 The STUKF method degenerates into the UKF method. The AWSTUKF method uses the adaptive method to calculate the weight of the measurement information to improve the sensitivity of the system. At $t_k = 303$ s, the small pulse maneuver was detected successfully, the fading factor rapidly enlarges the covariance matrix, therefore the filter converges quickly.

The computational costs of the three methods were estimated by 50 simulations. A window size of 20 s for the variance statistics function was set for each simulation. For the proposed method, the average run time was 4.12 s. The run time for the STUKF method and the UKF method was 3.93 and 3.22 s, respectively.

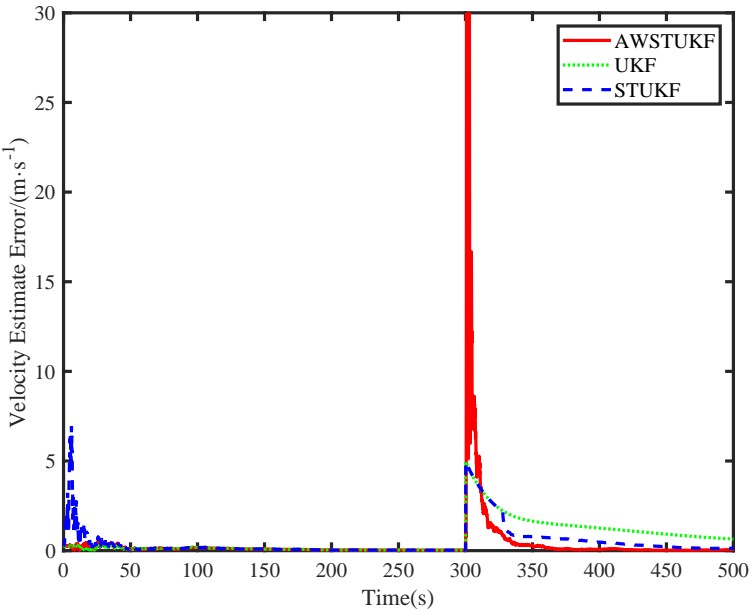

**Figure 4.** Velocity estimate errors of different methods.

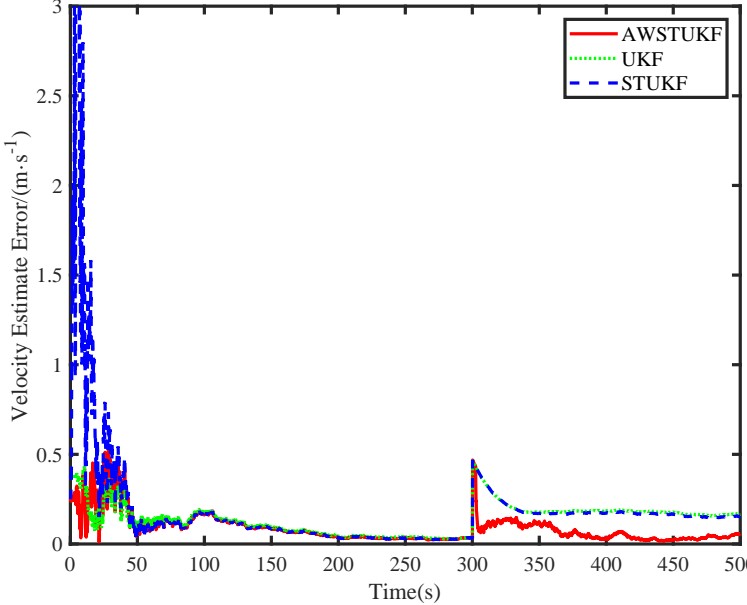

**Figure 5.** Position estimate errors of different methods.

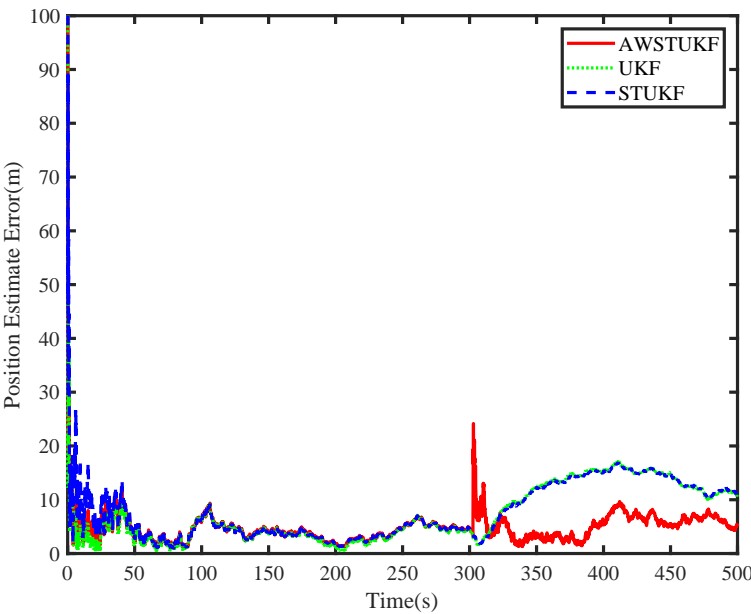

**Figure 6.** Velocity estimate errors of different methods.

The residual variation of the proposed AWSTUKF method and the STUKF in case 2 are presented in Figures 7 and 8. It can be observed that the residuals of ranging and angle measurement of the two methods have no large changes before and after the maneuver. The range rate increases rapidly when the maneuver occurs and it begins to converge after the maneuver. This showed that the residual orthogonality of the two methods changes after the maneuver, and the filtering gain needs to be changed by adjusting the fading factor. The orthogonality of the residual sequence also needs to be ensured. Compared with the STUKF method, because the weight is calculated in real-time, the convergence is achieved faster in the AWSTUKF method.

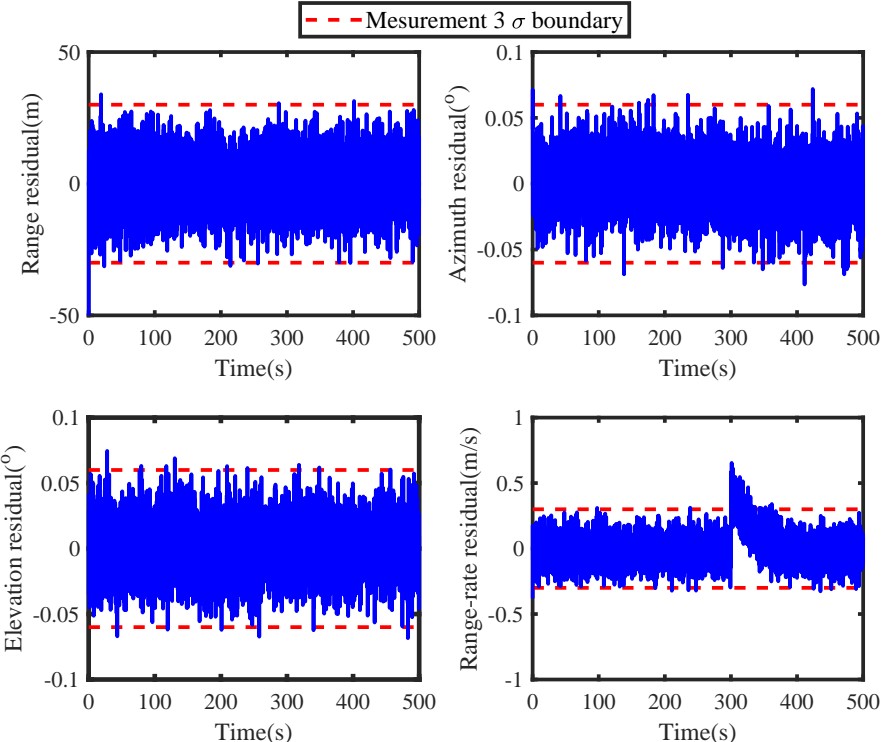

**Figure 7.** Measurement Residuals of the STUKF.

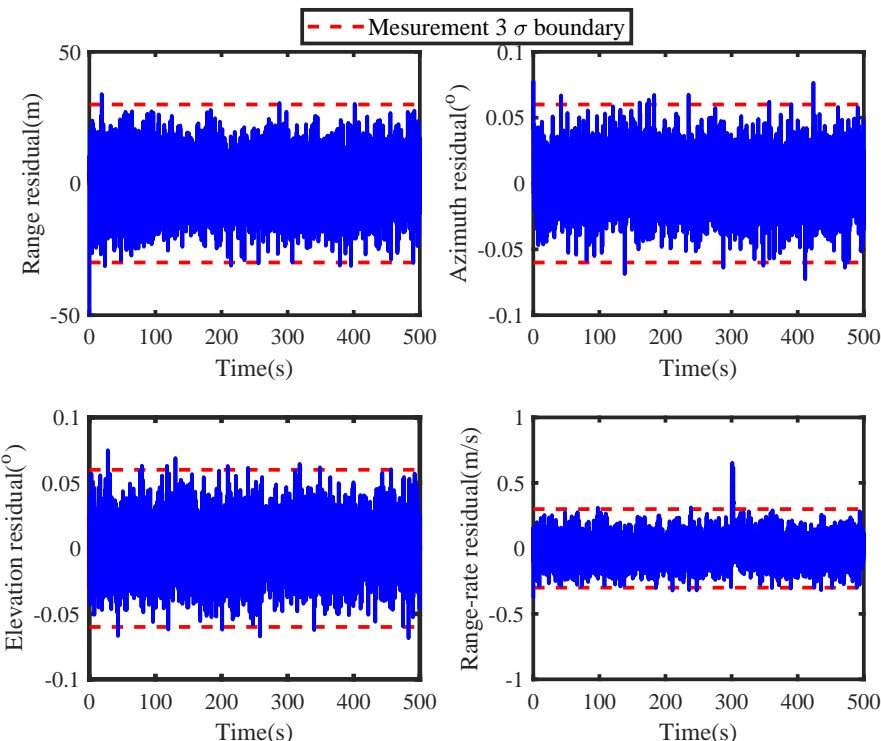

**Figure 8.** Measurement Residuals of the AWSTUKF.

The fading factors of the two methods in case 2 are shown in Figures 9 and 10. When the fading factor is greater than 1, the filter detects unknown maneuvering. the covariance matrix is amplified by the fading factor. It can be observed that in the TSTUKF method, the fading factor does not change significantly for a small pulse maneuver. For the proposed method, the fading factor exceeds the maneuvering detection threshold at 303 s and the detection delay is 3 s. This comparison shows that the proposed AWSTUKF method can detect the occurrence of unknown maneuvers quicker and more accurately.

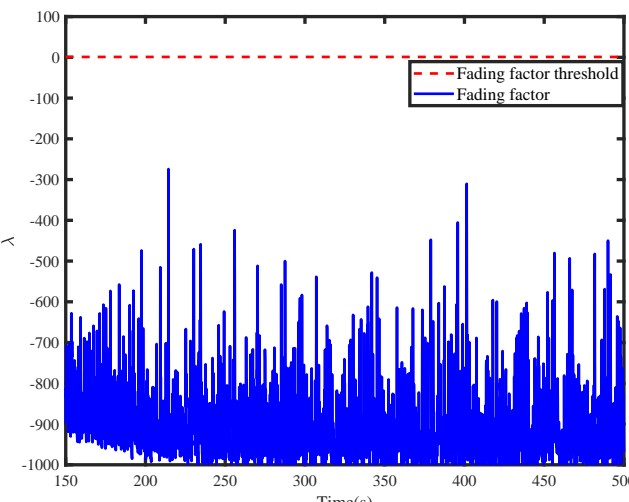

**Figure 9.** Fading Factors of the STUKF.

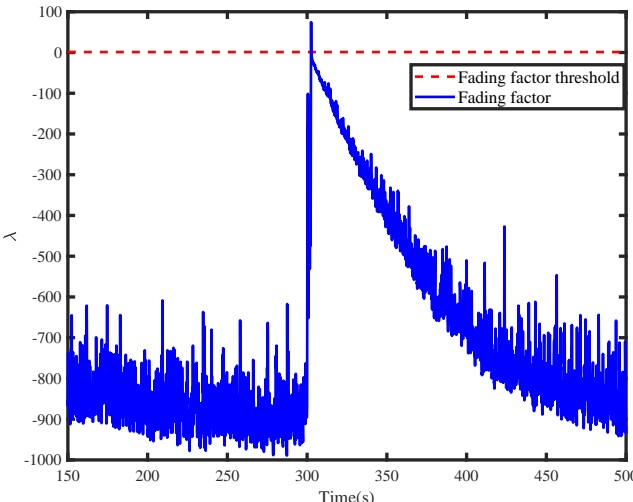

**Figure 10.** Fading Factors of the AWSTUKF.

Different magnitudes of impulsive maneuvers are used in a series of Monte Carlo simulations to demonstrate the superiorities of the AWSTUKF method over the STF method, the STUKF method, and the AWSTF method. The impulsive maneuvers are set at 0.5, 2.0, and 5.0 m/s in three groups of Monte Carlo simulations, and the other conditions are the same as in the previous case. Table 1 shows the root mean square errors (RMSES) of the estimations for the position and velocity after the impulsive maneuver ($t_k \geq 300$ s), and the detection delay of different methods. It can be seen that the AWSTUKF method outperforms the other three methods in terms of the position and the velocity estimation errors, and the detection delay. Both the the AWSTUKF and the AWSTF methods can successfully detect an impulsive maneuver when its magnitude is 0.5 m/s.

**Table 1.** Tracking performance of different methods in different situations.

| Parameters | STF | STUKF | AWSTF | AWSTUKF |
|---|---|---|---|---|
| | | **0.5 m/s** | | |
| **Position RMSE, m** | 13.15 | 12.09 | 6.19 | 5.39 |
| **Velocity RMSE, m/s** | 0.21 | 0.18 | 0.09 | 0.06 |
| **Detection delay, s** | none | none | 4.2 | 3.1 |
| | | **2.0 m/s** | | |
| **Position RMSE** | 38.23 | 37.18 | 6.96 | 6.56 |
| **Velocity RMSE, m/s** | 0.56 | 0.51 | 0.18 | 0.13 |
| **Detection delay, s** | 34.5 | 33.8 | 1.1 | 0.5 |
| | | **5.0 m/s** | | |
| **Position RMSE, m** | 41.46 | 40.24 | 8.14 | 7.44 |
| **Velocity RMSE, m/s** | 0.71 | 0.66 | 0.37 | 0.27 |
| **Detection delay, s** | 29.3 | 27.8 | 0.5 | 0.3 |

## 4. Conclusions

For relative navigation along long-range non-cooperative maneuvering targets, an adaptive weighted strong tracking unscented Kalman filter method was proposed. High-precision relative motion equation was employed for the target that is in medium and long distance. The adaptive variance function was designed to adjust the weight of the residual component in real-time, which enhances the sensitivity and robustness of fading factor to small pulse maneuvers. Simulations were carried out and the results showed that under different small pulse maneuvers, the AWSTUKF method has higher tracking accuracies and shorter maneuver detection delays compared with the STF, the AWSTF, and the STUKF methods. In future, the proposed method can be introduced into a space-borne platform

for detecting unknown maneuvers of a target spacecraft. Because of the sensitivity and robustness of the method, the platform would have a better tracking performance.

**Author Contributions:** All authors were contributed equally. Methodology, P.H. and H.L.; software, P.H., H.L., G.W., and Z.W.; validation, H.L., and G.W.; writing—review and editing, P.H., H.L., G.W., and Z.W.; funding acquisition, Z.W. and G.W.; supervision, Z.W. All authors have read and agreed to the published version of the manuscript.

**Funding:** This work was supported by the National Natural Science Foundation of China (grant numbers 11872034 and U20B2056), Manned Space Engineering Project of China (grant number 010201), and Natural Science Foundation of Beijing Municipal (grant number 1224039).

**Institutional Review Board Statement:** Not applicable.

**Informed Consent Statement:** Not applicable.

**Data Availability Statement:** The datasets employed during the current study are available from the corresponding author on request.

**Conflicts of Interest:** The authors declare no conflict of interest.

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
