# Peer review of "Application of Adaptive Weighted Strong Tracking Unscented Kalman Filter in Non-Cooperative Maneuvering Target Tracking"

_aerospace, doi:10.3390/aerospace9080468_

Round 1
Reviewer 1 Report
The theme of the article is relevant, the material is presented clearly and logical. I believe that the article can be published without improvements.
Author Response
Thank you very much for your comments.
Reviewer 2 Report
This is an interesting paper about the modification of unscented Kalman filter to solve the problem of non cooperative maneuvering target tracking
The following remarks should be taken into account:
-Your propose is limited to a Gaussian assumption. Thus, how can you guarantee that such assumption is satisfied?
-What do you mean in your sentence: "In this way, the sensitivity of the system regrading change of pulse maneuver is improved". According to Merriam Webster Dictionary, the verb regrade means: "to assign a new grade or mark to (something)" Did you mean "regarding"?
-In simulation sections, you are comparing the performance of your algorithm with respecto to another strategies. Some words about the computational cost of the three methods should be added.
-You should consider future work in your conclusion section, for example, what other kind of problems could be solved by using your algorithm?
Reviewer 3 Report
The article considers the interesting problem of tracking the space moving targets. The problem is generally nonlinear and the target dynamics is written in full details. However even in the dynamics description the possible perturbations are not clearly written and defined. The same remark relates to the observation process, in the article one needs to clearly define what can be observed and what kind of perturbations occur. So the implementation of EKF looks possible, but it needs adjustment which authors relates as adaptation. However, to verify the adaptation success one needs to test various approaches to adaptation and verification of the adaptation approach at final test.
Round 2
Reviewer 3 Report
In the new version of the article, the authors do not respond to the main comment of the review. What is the idea of the adaptation? As far as I understood, the authors try to keep the filtering and evaluation residuals orthogonal. This might help, but it is not a definitive approach to optimization and does not guarantee the best result. To ensure the reasonableness of the approach, it would be nice to simply compare the non-adaptive and "adaptive" approaches, just to demonstrate the effectiveness and benefits of their approach. Without such a comparison, I don't think the article deserves publication.
